# Efficient Non-greedy Optimization of Decision Trees

**Mohammad Norouzi**[1*]       **Maxwell D. Collins**[2*]       **Matthew Johnson**[3]
**David J. Fleet**[4]       **Pushmeet Kohli**[5]

[1,4] Department of Computer Science, University of Toronto
[2] Department of Computer Science, University of Wisconsin-Madison
[3,5] Microsoft Research

## Abstract

Decision trees and randomized forests are widely used in computer vision and machine learning. Standard algorithms for decision tree induction optimize the split functions one node at a time according to some splitting criteria. This greedy procedure often leads to suboptimal trees. In this paper, we present an algorithm for optimizing the split functions at all levels of the tree jointly with the leaf parameters, based on a global objective. We show that the problem of finding optimal linear-combination (oblique) splits for decision trees is related to structured prediction with latent variables, and we formulate a convex-concave upper bound on the tree's empirical loss. Computing the gradient of the proposed surrogate objective with respect to each training exemplar is $O(d^2)$, where $d$ is the tree depth, and thus training deep trees is feasible. The use of stochastic gradient descent for optimization enables effective training with large datasets. Experiments on several classification benchmarks demonstrate that the resulting non-greedy decision trees outperform greedy decision tree baselines.

## 1 Introduction

Decision trees and forests [5, 21, 4] have a long and rich history in machine learning [10, 7]. Recent years have seen an increase in their popularity, owing to their computational efficiency and applicability to large-scale classification and regression tasks. A case in point is Microsoft Kinect where decision trees are trained on millions of exemplars to enable real-time human pose estimation from depth images [22].

Conventional algorithms for decision tree induction are greedy. They grow a tree one node at a time following procedures laid out decades ago by frameworks such as ID3 [21] and CART [5]. While recent work has proposed new objective functions to guide greedy algorithms [20, 12], it continues to be the case that decision tree applications (*e.g.,* [9, 14]) utilize the same dated methods of tree induction. Greedy decision tree induction builds a binary tree via a recursive procedure as follows: beginning with a single node, indexed by $i$, a split function $s_i$ is optimized based on a corresponding subset of the training data $\mathcal{D}_i$ such that $\mathcal{D}_i$ is split into two subsets, which in turn define the training data for the two children of the node $i$. The intrinsic limitation of this procedure is that the optimization of $s_i$ is solely conditioned on $\mathcal{D}_i$, *i.e.,* there is no ability to fine-tune the split function $s_i$ based on the results of training at lower levels of the tree. This paper proposes a general framework for non-greedy learning of the split parameters for tree-based methods that addresses this limitation. We focus on binary trees, while extension to $N$-ary trees is possible. We show that our joint optimization of the split functions at different levels of the tree under a global objective not only promotes cooperation between the split nodes to create more compact trees, but also leads to better generalization performance.

---

[*]Part of this work was done while M. Norouzi and M. D. Collins were at Microsoft Research, Cambridge.

One of the key contributions of this work is establishing a link between the decision tree optimization problem and the problem of structured prediction with latent variables [25]. We present a novel formulation of the decision tree learning that associates a binary latent decision variable with each split node in the tree and uses such latent variables to formulate the tree's empirical loss. Inspired by advances in structured prediction [23, 24, 25], we propose a convex-concave upper bound on the empirical loss. This bound acts as a surrogate objective that is optimized using stochastic gradient descent (SGD) to find a locally optimal configuration of the split functions. One complication introduced by this particular formulation is that the number of latent decision variables grows exponentially with the tree depth $d$. As a consequence, each gradient update will have a complexity of $O(2^d p)$ for $p$-dimensional inputs. One of our technical contributions is showing how this complexity can be reduced to $O(d^2 p)$ by modifying the surrogate objective, thereby enabling efficient learning of deep trees.

## 2   Related work

Finding optimal split functions at different levels of a decision tree according to some global objective, such as a regularized empirical risk, is NP-complete [11] due to the discrete and sequential nature of the decisions in a tree. Thus, finding an efficient alternative to the greedy approach has remained a difficult objective despite many prior attempts.

Bennett [1] proposes a non-greedy multi-linear programming based approach for global tree optimization and shows that the method produces trees that have higher classification accuracy than standard greedy trees. However, their method is limited to binary classification with 0-1 loss and has a high computation complexity, making it only applicable to trees with few nodes.

The work in [15] proposes a means for training decision forests in an online setting by incrementally extending the trees as new data points are added. As opposed to a naive incremental growing of the trees, this work models the decision trees with Mondrian Processes.

The Hierarchical Mixture of Experts model [13] uses soft splits rather than hard binary decisions to capture situations where the transition from low to high response is gradual. The use of soft splits at internal nodes of the tree yields a probabilistic model in which the log-likelihood is a smooth function of the unknown parameters. Hence, training based on log-likelihood is amenable to numerical optimization via methods such as expectation maximization (EM). That said, the soft splits necessitate the evaluation of all or most of the experts for each data point, so much of the computational advantage of the decision tree are lost.

Murthy and Salzburg [17] argue that non-greedy tree learning methods that work by looking ahead are unnecessary and sometimes harmful. This is understandable since their methods work by minimizing the empirical loss without any regularization, which is prone to overfitting. To avoid this problem, it is a common practice (see Breiman [4] or Criminisi and Shotton [7] for an overview) to limit the tree depth and introduce limits on the number of training instances below which a tree branch is not extended, or to force a diverse ensemble of trees (*i.e.,* a decision forest) through the use of bagging. Bennett and Blue [2] describe a different way to overcome overfitting by using max-margin framework and the Support Vector Machines (SVM) at the split nodes of the tree. Subsequently, Bennett *et al.* [3] show how enlarging the margin of decision tree classifiers results in better generalization performance.

Our formulation for decision tree induction improves on prior art in a number of ways. Not only does our latent variable formulation of decision trees enable efficient learning, it can handle any general loss function while not sacrificing the $O(dp)$ complexity of inference imparted by the tree structure. Further, our surrogate objective provides a natural way to regularize the joint optimization of tree parameters to discourage overfitting.

## 3   Problem formulation

For ease of exposition, this paper focuses on binary classification trees, with $m$ internal (split) nodes, and $m + 1$ leaf (terminal) nodes. Note that in a binary tree the number of leaves is always one more than the number of internal (non-leaf) nodes. An input, $\mathbf{x} \in \mathbb{R}^p$, is directed from the root of the tree down through internal nodes to a leaf node. Each leaf node specifies a distribution over $k$ class labels. Each internal node, indexed by $i \in \{1, \ldots, m\}$, performs a binary test by evaluating a node-

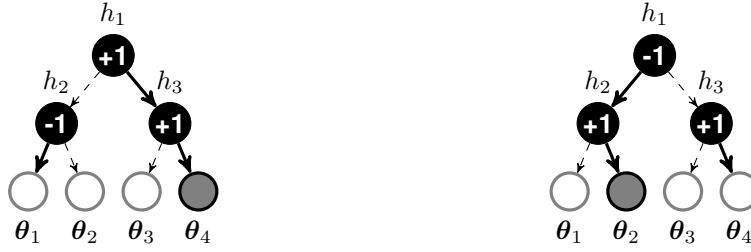

$$f([+1, -1, +1]^\mathsf{T}) = [0, 0, 0, 1]^\mathsf{T} = \mathbb{1}_4 \qquad\qquad f([-1, +1, +1]^\mathsf{T}) = [0, 1, 0, 0]^\mathsf{T} = \mathbb{1}_2$$

$$\boldsymbol{\theta} = \Theta^\mathsf{T} f(\mathbf{h}) = \boldsymbol{\theta}_4 \qquad\qquad\qquad \boldsymbol{\theta} = \Theta^\mathsf{T} f(\mathbf{h}) = \boldsymbol{\theta}_2$$

Figure 1: The binary split decisions in a decision tree with $m = 3$ internal nodes can be thought as a binary vector $\mathbf{h} = [h_1, h_2, h_3]^\mathsf{T}$. Tree navigation to reach a leaf can be expressed in terms of a function $f(\mathbf{h})$. The selected leaf parameters can be expressed by $\boldsymbol{\theta} = \Theta^\mathsf{T} f(\mathbf{h})$.

specific split function $s_i(\mathbf{x}) : \mathbb{R}^p \to \{-1, +1\}$. If $s_i(\mathbf{x})$ evaluates to $-1$, then $\mathbf{x}$ is directed to the left child of node $i$. Otherwise, $\mathbf{x}$ is directed to the right child. And so on down the tree. Each split function $s_i(\cdot)$, by parameterized a weight vector $\mathbf{w}_i$, is assumed to be a linear threshold function, *i.e.*, $s_i(\mathbf{x}) = \mathrm{sgn}(\mathbf{w}_i^\mathsf{T}\mathbf{x})$. We incorporate an offset parameter to obtain split functions of the form $\mathrm{sgn}(\mathbf{w}_i^\mathsf{T}\mathbf{x} - b_i)$ by appending a constant "$-1$" to the input feature vector.

Each leaf node, indexed by $j \in \{1, \ldots, m+1\}$, specifies a conditional probability distribution over class labels, $l \in \{1, \ldots, k\}$, denoted $p(y = l \mid j)$. Leaf distributions are parametrized with a vector of unnormalized predictive log-probabilities, denoted $\boldsymbol{\theta}_j \in \mathbb{R}^k$, and a softmax function; *i.e.*,

$$p(y = l \mid j) = \frac{\exp\{\boldsymbol{\theta}_{j[l]}\}}{\sum_{\alpha=1}^{k} \exp\{\boldsymbol{\theta}_{j[\alpha]}\}}, \tag{1}$$

where $\boldsymbol{\theta}_{j[\alpha]}$ denotes the $\alpha^{\text{th}}$ element of vector $\boldsymbol{\theta}_j$.

The parameters of the tree comprise the $m$ internal weight vectors, $\{\mathbf{w}_i\}_{i=1}^{m}$, and the $m+1$ vectors of unnormalized log-probabilities, one for each leaf node, $\{\boldsymbol{\theta}_j\}_{j=1}^{m+1}$. We pack these parameters into two matrices $W \in \mathbb{R}^{m \times p}$ and $\Theta \in \mathbb{R}^{(m+1) \times k}$ whose rows comprise weight vectors and leaf parameters, *i.e.*, $W \equiv [\mathbf{w}_1, \ldots, \mathbf{w}_m]^\mathsf{T}$ and $\Theta \equiv [\boldsymbol{\theta}_1, \ldots, \boldsymbol{\theta}_{m+1}]^\mathsf{T}$. Given a dataset of input-output pairs, $\mathcal{D} \equiv \{\mathbf{x}_z, y_z\}_{z=1}^{n}$, where $y_z \in \{1, \ldots, k\}$ is the ground truth class label associated with input $\mathbf{x}_z \in \mathbb{R}^p$, we wish to find a joint configuration of oblique splits $W$ and leaf parameters $\Theta$ that minimize some measure of misclassification loss on the training dataset. Joint optimization of the split functions and leaf parameters according to a global objective is known to be extremely challenging [11] due to the discrete and sequential nature of the splitting decisions within the tree.

One can evaluate all of the split functions, for every internal node of the tree, on input $\mathbf{x}$ by computing $\mathrm{sgn}(W\mathbf{x})$, where $\mathrm{sgn}(\cdot)$ is the element-wise sign function. One key idea that helps linking decision tree learning to latent structured prediction is to think of an $m$-bit vector of potential split decisions, *e.g.*, $\mathbf{h} = \mathrm{sgn}(W\mathbf{x}) \in \{-1, +1\}^m$, as a latent variable. Such a latent variable determines the leaf to which a data point is directed, and then classified using the leaf parameters. To formulate the loss for $(\mathbf{x}, y)$, we introduce a tree navigation function $f : \mathbb{H}^m \to \mathbb{I}_{m+1}$ that maps an $m$-bit sequence of split decisions ($\mathbb{H}^m \equiv \{-1, +1\}^m$) to an indicator vector that specifies a 1-of-$(m+1)$ encoding. Such an indicator vector is only non-zero at the index of the selected leaf. Fig. 1 illustrates the tree navigation function for a tree with 3 internal nodes.

Using the notation developed above, $\boldsymbol{\theta} = \Theta^\mathsf{T} f(\mathrm{sgn}(W\mathbf{x}))$ represents the parameters corresponding to the leaf to which $\mathbf{x}$ is directed by the split functions in $W$. A generic loss function of the form $\ell(\boldsymbol{\theta}, y)$ measures the discrepancy between the model prediction based on $\boldsymbol{\theta}$ and an output $y$. For the softmax model given by (1), a natural loss is the negative log probability of the correct label, referred to as *log loss*,

$$\ell(\boldsymbol{\theta}, y) = \ell_{\log}(\boldsymbol{\theta}, y) = -\boldsymbol{\theta}_{[y]} + \log\left(\sum_{\beta=1}^{k} \exp(\boldsymbol{\theta}_{[\beta]})\right). \tag{2}$$

For regression tasks, when $\mathbf{y} \in \mathbb{R}^q$, and the value of $\boldsymbol{\theta} \in \mathbb{R}^q$ is directly emitted as the model prediction, a natural choice of $\ell$ is squared loss,

$$\ell(\boldsymbol{\theta}, \mathbf{y}) = \ell_{\mathrm{sqr}}(\boldsymbol{\theta}, \mathbf{y}) = \|\boldsymbol{\theta} - \mathbf{y}\|^2 . \tag{3}$$

One can adopt other forms of loss within our decision tree learning framework as well. The goal of learning is to find $W$ and $\Theta$ that minimize empirical loss, for a given training set $\mathcal{D}$, that is,

$$\mathcal{L}(W, \Theta; \mathcal{D}) = \sum_{(\mathbf{x}, y) \in \mathcal{D}} \ell\left(\Theta^\mathsf{T} f(\mathrm{sgn}(W\mathbf{x})), y\right) . \tag{4}$$

Direct global optimization of empirical loss $\mathcal{L}(W, \Theta; \mathcal{D})$ with respect to $W$ is challenging. It is a discontinuous and piecewise-constant function of $W$. Furthermore, given an input $\mathbf{x}$, the navigation function $f(\cdot)$ yields a leaf parameter vector based on a sequence of binary tests, where the results of the initial tests determine which subsequent tests are performed. It is not clear how this dependence of binary tests should be formulated.

## 4  Decision trees and structured prediction

To overcome the intractability in the optimization of $\mathcal{L}$, we develop a piecewise smooth upper bound on empirical loss. Our upper bound is inspired by the formulation of structured prediction with latent variables [25]. A key observation that links decision tree learning to structured prediction, is that one can re-express $\mathrm{sgn}(W\mathbf{x})$ in terms of a latent variable $\mathbf{h}$. That is,

$$\mathrm{sgn}(W\mathbf{x}) = \operatorname*{argmax}_{\mathbf{h} \in \mathbb{H}^m}(\mathbf{h}^\mathsf{T} W\mathbf{x}) . \tag{5}$$

In this form, decision tree's split functions implicitly map an input $\mathbf{x}$ to a binary vector $\mathbf{h}$ by maximizing a score function $\mathbf{h}^\mathsf{T} W\mathbf{x}$, the inner product of $\mathbf{h}$ and $W\mathbf{x}$. One can re-express the score function in terms of a more familiar form of a joint feature space on $\mathbf{h}$ and $\mathbf{x}$, as $\mathbf{w}^\mathsf{T} \phi(\mathbf{h}, \mathbf{x})$, where $\phi(\mathbf{h}, \mathbf{x}) = \mathrm{vec}\left(\mathbf{h}\mathbf{x}^\mathsf{T}\right)$, and $\mathbf{w} = \mathrm{vec}\left(W\right)$. Previously, Norouzi and Fleet [19] used the same reformulation (5) of linear threshold functions to learn binary similarity preserving hash functions.

Given (5), we re-express empirical loss as,

$$\mathcal{L}(W, \Theta; \mathcal{D}) = \sum_{(\mathbf{x}, y) \in \mathcal{D}} \ell(\Theta^\mathsf{T} f(\widehat{\mathbf{h}}(\mathbf{x})), y) ,$$
$$where \quad \widehat{\mathbf{h}}(\mathbf{x}) = \operatorname*{argmax}_{\mathbf{h} \in \mathbb{H}^m}(\mathbf{h}^\mathsf{T} W\mathbf{x}) . \tag{6}$$

This objective resembles the objective functions used in structured prediction, and since we do not have *a priori* access to the ground truth split decisions, $\widehat{\mathbf{h}}(\mathbf{x})$, this problem is a form of structured prediction with latent variables.

## 5  Upper bound on empirical loss

We develop an upper bound on loss for an input-output pair $(\mathbf{x}, y)$, which takes the form,

$$\ell(\Theta^\mathsf{T} f(\mathrm{sgn}(W\mathbf{x})), y) \leq \max_{\mathbf{g} \in \mathbb{H}^m}\left(\mathbf{g}^\mathsf{T} W\mathbf{x} + \ell(\Theta^\mathsf{T} f(\mathbf{g}), y)\right) - \max_{\mathbf{h} \in \mathbb{H}^m}(\mathbf{h}^\mathsf{T} W\mathbf{x}) . \tag{7}$$

To validate the bound, first note that the second term on the RHS is maximized by $\mathbf{h} = \widehat{\mathbf{h}}(\mathbf{x}) = \mathrm{sgn}(W\mathbf{x})$. Second, when $\mathbf{g} = \widehat{\mathbf{h}}(\mathbf{x})$, it is clear that the LHS equals the RHS. Finally, for all other values of $\mathbf{g}$, the RHS can only get larger than when $\mathbf{g} = \widehat{\mathbf{h}}(\mathbf{x})$ because of the max operator. Hence, the inequality holds. An algebraic proof of (7) is presented in the supplementary material.

In the context of structured prediction, the first term of the upper bound, *i.e.,* the maximization over $\mathbf{g}$, is called *loss-augmented inference*, as it augments the inference problem, *i.e.,* the maximization over $\mathbf{h}$, with a loss term. Fortunately, the loss-augmented inference for our decision tree learning formulation can be solved exactly, as discussed below.

It is also notable that the loss term on the LHS of (7) is invariant to the scale of $W$, but the upper bound on the right side of (7) is not. As a consequence, as with binary SVM and margin-rescaling formulations of structural SVM [24], we introduce a regularizer on the norm of $W$ when optimizing the bound. To justify the regularizer, we discuss the effect of the scale of $W$ on the bound.

**Proposition 1.** *The upper bound on the loss becomes tighter as a constant multiple of $W$ increases, i.e., for $a > b > 0$:*

$$\max_{\mathbf{g} \in \mathbb{H}^m} \left( a\mathbf{g}^\mathsf{T} W\mathbf{x} + \ell(\Theta^\mathsf{T} f(\mathbf{g}), y) \right) - \max_{\mathbf{h} \in \mathbb{H}^m} (a\mathbf{h}^\mathsf{T} W\mathbf{x}) \leq$$
$$\max_{\mathbf{g} \in \mathbb{H}^m} \left( b\mathbf{g}^\mathsf{T} W\mathbf{x} + \ell(\Theta^\mathsf{T} f(\mathbf{g}), y) \right) - \max_{\mathbf{h} \in \mathbb{H}^m} (b\mathbf{h}^\mathsf{T} W\mathbf{x}). \tag{8}$$

*Proof.* Please refer to the supplementary material for the proof. $\qquad\square$

In the limit, as the scale of $W$ approach $+\infty$, the loss term $\ell(\Theta^\mathsf{T} f(\mathbf{g}), y)$ becomes negligible compared to the score term $\mathbf{g}^\mathsf{T} W\mathbf{x}$. Thus, the solutions to loss-augmented inference and inference problems become almost identical, except when an element of $W\mathbf{x}$ is very close to $0$. Thus, even though a larger $\|W\|$ yields a tighter bound, it makes the bound approach the loss itself, and therefore becomes nearly piecewise-constant, which is hard to optimize. Based on Proposition 1, one easy way to decrease the upper bound is to increase the norm of $W$, which does not affect the loss.

Our experiments indicate that a lower value of the loss can be achieved when the norm of $W$ is regularized. We therefore constrain the norm of $W$ to obtain an objective with better generalization. Since each row of $W$ acts independently in a decision tree in the split functions, it is reasonable to constrain the norm of each row independently. Summing over the bounds for different training pairs and constraining the norm of rows of $W$, we obtain the following optimization problem, called the *surrogate* objective:

$$\text{minimize } \mathcal{L}'(W, \Theta; \mathcal{D}) = \sum_{(\mathbf{x}, y) \in \mathcal{D}} \left( \max_{\mathbf{g} \in \mathbb{H}^m} \left( \mathbf{g}^\mathsf{T} W\mathbf{x} + \ell(\Theta^\mathsf{T} f(\mathbf{g}), y) \right) - \max_{\mathbf{h} \in \mathbb{H}^m} (\mathbf{h}^\mathsf{T} W\mathbf{x}) \right) \tag{9}$$
$$s.t. \quad \|\mathbf{w}_i\|^2 \leq \nu \quad \text{for all } i \in \{1, \dots, m\}$$

where $\nu \in \mathbb{R}^+$ is a regularization parameter and $\mathbf{w}_i$ is the $i^\text{th}$ row of $W$. For all values of $\nu$, we have $\mathcal{L}(W, \Theta; \mathcal{D}) \leq \mathcal{L}'(W, \Theta; \mathcal{D})$. Instead of using the typical Lagrange form for regularization, we employ hard constraints to enable sparse gradient updates of the rows of $W$, since the gradients for most rows of $W$ are zero at each step in training.

## 6 Optimizing the surrogate objective

Even though minimizing the surrogate objective of (9) entails a non-convex optimization, $\mathcal{L}'(W, \Theta; \mathcal{D})$ is much better behaved than empirical loss in (4). $\mathcal{L}'(W, \Theta; \mathcal{D})$ is piecewise linear and convex-concave in $W$, and the constraints on $W$ define a convex set.

**Loss-augmented inference.** To evaluate and use the surrogate objective in (9) for optimization, we must solve a *loss-augmented inference* problem to find the binary code that maximizes the sum of the score and loss terms:

$$\widehat{\mathbf{g}}(\mathbf{x}) = \operatorname*{argmax}_{\mathbf{g} \in \mathbb{H}^m} \left( \mathbf{g}^\mathsf{T} W\mathbf{x} + \ell(\Theta^\mathsf{T} f(\mathbf{g}), y) \right). \tag{10}$$

An observation that makes this optimization tractable is that $f(\mathbf{g})$ can only take on $m+1$ distinct values, which correspond to terminating at one of the $m+1$ leaves of the tree and selecting a leaf parameter from $\{\boldsymbol{\theta}_j\}_{j=1}^{m+1}$. Fortunately, for any leaf index $j \in \{1, \dots, m+1\}$, we can solve

$$\operatorname*{argmax}_{\mathbf{g} \in \mathbb{H}^m} \left( \mathbf{g}^\mathsf{T} W\mathbf{x} + \ell(\boldsymbol{\theta}_j, y) \right) \quad s.t. \quad f(\mathbf{g}) = \mathbb{1}_j, \tag{11}$$

efficiently. Note that if $f(\mathbf{g}) = \mathbb{1}_j$, then $\Theta^\mathsf{T} f(\mathbf{g})$ equals the $j^\text{th}$ row of $\Theta$, *i.e.,* $\boldsymbol{\theta}_j$. To solve (11) we need to set all of the binary bits in $\mathbf{g}$ corresponding to the path from the root to the leaf $j$ to be consistent with the path direction toward the leaf $j$. However, bits of $\mathbf{g}$ that do not appear on this path have no effect on the output of $f(\mathbf{g})$, and all such bits should be set based on $\mathbf{g}_{[i]} = \operatorname{sgn}(\mathbf{w}_i^\mathsf{T} \mathbf{x})$ to obtain maximum $\mathbf{g}^\mathsf{T} W\mathbf{x}$. Accordingly, we can essentially ignore the off-the-path bits by subtracting $\operatorname{sgn}(W\mathbf{x})^\mathsf{T} W\mathbf{x}$ from (11) to obtain,

$$\operatorname*{argmax}_{\mathbf{g} \in \mathbb{H}^m} \left( \mathbf{g}^\mathsf{T} W\mathbf{x} + \ell(\boldsymbol{\theta}_j, y) \right) = \operatorname*{argmax}_{\mathbf{g} \in \mathbb{H}^m} \left( (\mathbf{g} - \operatorname{sgn}(W\mathbf{x}))^\mathsf{T} W\mathbf{x} + \ell(\boldsymbol{\theta}_j, y) \right). \tag{12}$$

---
**Algorithm 1** Stochastic gradient descent (SGD) algorithm for non-greedy decision tree learning.
---
1: Initialize $W^{(0)}$ and $\Theta^{(0)}$ using greedy procedure
2: **for** $t = 0$ to $\tau$ **do**
3:     Sample a pair $(\mathbf{x}, y)$ uniformly at random from $\mathcal{D}$
4:     $\widehat{\mathbf{h}} \leftarrow \mathrm{sgn}(W^{(t)}\mathbf{x})$
5:     $\widehat{\mathbf{g}} \leftarrow \mathrm{argmax}_{\mathbf{g} \in \mathcal{H}^m} \left\{ \mathbf{g}^{\mathsf{T}} W^{(t)} \mathbf{x} + \ell(\Theta^{\mathsf{T}} f(\mathbf{g}), y) \right\}$
6:     $W^{(tmp)} \leftarrow W^{(t)} - \eta\, \widehat{\mathbf{g}}\mathbf{x}^{\mathsf{T}} + \eta\, \widehat{\mathbf{h}}\mathbf{x}^{\mathsf{T}}$
7:     **for** $i = 1$ to $m$ **do**
8:         $W_{i,.}^{(t+1)} \leftarrow \min\left\{1, \sqrt{\nu}\big/\left\|W_{i,.}^{(tmp)}\right\|_2\right\} W_{i,.}^{(tmp)}$
9:     **end for**
10:    $\Theta^{(t+1)} \leftarrow \Theta^{(t)} - \eta\, \frac{\partial}{\partial \Theta} \ell(\Theta^{\mathsf{T}} f(\widehat{\mathbf{g}}), y)\big|_{\Theta = \Theta^{(t)}}$
11: **end for**
---

Note that $\mathrm{sgn}(W\mathbf{x})^{\mathsf{T}} W\mathbf{x}$ is constant in $\mathbf{g}$, and this subtraction zeros out all bits in $\mathbf{g}$ that are not on the path to the leaf $j$. So, to solve (12), we only need to consider the bits on the path to the leaf $j$ for which $\mathrm{sgn}(\mathbf{w}_i{}^{\mathsf{T}}\mathbf{x})$ is not consistent with the path direction. Using a single depth-first search on the decision tree, we can solve (11) for every $j$, and among those, we pick the one that maximizes (11).

The algorithm described above is $O(mp) \subseteq O(2^d p)$, where $d$ is the tree depth, and we require a multiple of $p$ for computing the inner product $\mathbf{w}_i\mathbf{x}$ at each internal node $i$. This algorithm is not efficient for deep trees, especially as we need to perform loss-augmented inference once for every stochastic gradient computation. In what follows, we develop an alternative more efficient formulation and algorithm with time complexity of $O(d^2 p)$.

**Fast loss-augmented inference.** To motivate the fast loss-augmented inference algorithm, we formulate a slightly different upper bound on the loss, *i.e.,*

$$\ell(\Theta^{\mathsf{T}} f(\mathrm{sgn}(W\mathbf{x})), y) \ \leq\ \max_{\mathbf{g} \in \mathcal{B}_1(\mathrm{sgn}(W\mathbf{x}))} \left(\mathbf{g}^{\mathsf{T}} W\mathbf{x} + \ell(\Theta^{\mathsf{T}} f(\mathbf{g}), y)\right) - \max_{\mathbf{h} \in \mathbb{H}^m} \left(\mathbf{h}^{\mathsf{T}} W\mathbf{x}\right), \qquad (13)$$

where $\mathcal{B}_1(\mathrm{sgn}(W\mathbf{x}))$ denotes the Hamming ball of radius 1 around $\mathrm{sgn}(W\mathbf{x})$, *i.e.,* $\mathcal{B}_1(\mathrm{sgn}(W\mathbf{x})) \equiv \{\mathbf{g} \in \mathbb{H}^m \mid \|\mathbf{g} - \mathrm{sgn}(W\mathbf{x})\|_H \leq 1\}$, hence $\mathbf{g} \in \mathcal{B}_1(\mathrm{sgn}(W\mathbf{x}))$ implies that $\mathbf{g}$ and $\mathrm{sgn}(W\mathbf{x})$ differ in at most one bit. The proof of (13) is identical to the proof of (7). The key benefit of this new formulation is that loss-augmented inference with the new bound is computationally efficient. Since $\widehat{\mathbf{g}}$ and $\mathrm{sgn}(W\mathbf{x})$ differ in at most one bit, then $f(\widehat{\mathbf{g}})$ can only take $d + 1$ distinct values. Thus we need to evaluate (12) for at most $d + 1$ values of $j$, requiring a running time of $O(d^2 p)$.

**Stochastic gradient descent (SGD).** One reasonable approach to minimizing (9) uses stochastic gradient descent (SGD), the steps of which are outlined in Alg 1. Here, $\eta$ denotes the learning rate, and $\tau$ is the number of optimization steps. Line 6 corresponds to a gradient update in $W$, which is supported by the fact that $\frac{\partial}{\partial W} \mathbf{h}^{\mathsf{T}} W\mathbf{x} = \mathbf{h}\mathbf{x}^{\mathsf{T}}$. Line 8 performs projection back to the feasible region of $W$, and Line 10 updates $\Theta$ based on the gradient of loss. Our implementation modifies Alg 1 by adopting common SGD tricks, including the use of momentum and mini-batches.

**Stable SGD (SSGD).** Even though Alg 1 achieves good training and test accuracy relatively quickly, we observe that after several gradient updates some of the leaves may end up not being assigned to any data points and hence the full tree capacity is not exploited. We call such leaves *inactive* as opposed to *active* leaves that are assigned to at least one training data point. An inactive leaf may become active again, but this rarely happens given the form of gradient updates. To discourage abrupt changes in the number of inactive leaves, we introduce a variant of SGD, in which the assignments of data points to leaves are fixed for a number of gradient updates. Thus, the bound is optimized with respect to a set of data point leaf assignment constraints. When the improvement in the bound becomes negligible the leaf assignment variables are updated, followed by another round of optimization of the bound. We call this algorithm *Stable SGD (SSGD)* because it changes the assignment of data points to leaves more conservatively than SGD. Let $a(\mathbf{x})$ denote the 1-of-$(m + 1)$ encoding of the leaf to which a data point $\mathbf{x}$ should be assigned to. Then, each iteration of SSGD

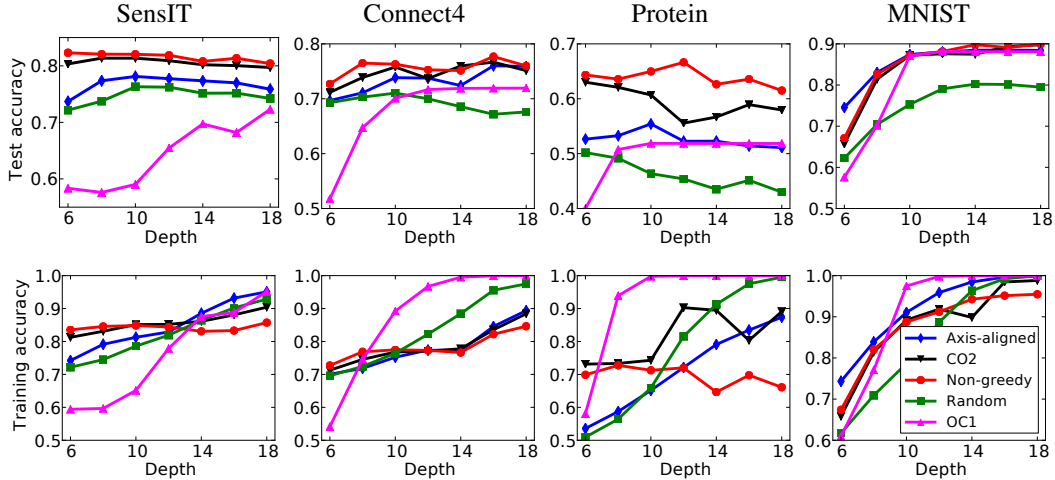

Figure 2: Test and training accuracy of a single tree as a function of tree depth for different methods. Non-greedy trees achieve better test accuracy throughout different depths. Non-greedy exhibit less vulnerability to overfitting.

with fast loss-augmented inference relies on the following upper bound on loss,

$$\ell(\Theta^\mathsf{T} f(\mathrm{sgn}(W\mathbf{x})), y) \leq \max_{\mathbf{g} \in \mathcal{B}_1(\mathrm{sgn}(W\mathbf{x}))} \big( \mathbf{g}^\mathsf{T} W\mathbf{x} + \ell(\Theta^\mathsf{T} f(\mathbf{g}), y) \big) - \max_{\mathbf{h} \in \mathbb{H}^m | f(\mathbf{h}) = a(\mathbf{x})} \big( \mathbf{h}^\mathsf{T} W\mathbf{x} \big) .$$

(14)

One can easily verify that the RHS of (14) is larger than the RHS of (13), hence the inequality.

**Computational complexity.** To analyze the computational complexity of each SGD step, we note that Hamming distance between $\widehat{\mathbf{g}}$ (defined in (10)) and $\widehat{\mathbf{h}} = \mathrm{sgn}(W\mathbf{x})$ is bounded above by the depth of the tree $d$. This is because only those elements of $\widehat{\mathbf{g}}$ corresponding to the path to a selected leaf can differ from $\mathrm{sgn}(W\mathbf{x})$. Thus, for SGD the expression $(\widehat{\mathbf{g}} - \widehat{\mathbf{h}})\mathbf{x}^\mathsf{T}$ needed for Line 6 of Alg 1 can be computed in $O(dp)$, if we know which bits of $\widehat{\mathbf{h}}$ and $\widehat{\mathbf{g}}$ differ. Accordingly, Lines 6 and 7 can be performed in $O(dp)$. The computational bottleneck is the loss augmented inference in Line 5. When fast loss-augmented inference is performed in $O(d^2 p)$ time, the total time complexity of gradient update for both SGD and SSGD becomes $O(d^2 p + k)$, where $k$ is the number of labels.

## 7 Experiments

Experiments are conducted on several benchmark datasets from LibSVM [6] for multi-class classification, namely *SensIT*, *Connect4*, *Protein*, and *MNIST*. We use the provided train; validation; test sets when available. If such splits are not provided, we use a random $80\%/20\%$ split of the training data for train; validation and a random $64\%/16\%/20\%$ split for train; validation; test sets.

We compare our method for *non-greedy* learning of oblique trees with several greedy baselines, including conventional *axis-aligned* trees based on information gain, *OC1* oblique trees [17] that use coordinate descent for optimization of the splits, and *random* oblique trees that select the best split function from a set of randomly generated hyperplanes based on information gain. We also compare with the results of *CO2* [18], which is a special case of our upper bound approach applied greedily to trees of depth $1$, one node at a time. Any base algorithm for learning decision trees can be augmented by post-training pruning [16], or building ensembles with bagging [4] or boosting [8]. However, the key differences between non-greedy trees and baseline greedy trees become most apparent when analyzing individual trees. For a single tree the major determinant of accuracy is the size of the tree, which we control by changing the maximum tree depth.

Fig. 2 depicts test and training accuracy for non-greedy trees and four other baselines as function of tree depth. We evaluate trees of depth $6$ up to $18$ at depth intervals of $2$. The hyper-parameters for each method are tuned for each depth independently. While the absolute accuracy of our non-greedy trees varies between datasets, a few key observations hold for all cases. First, we observe that non-

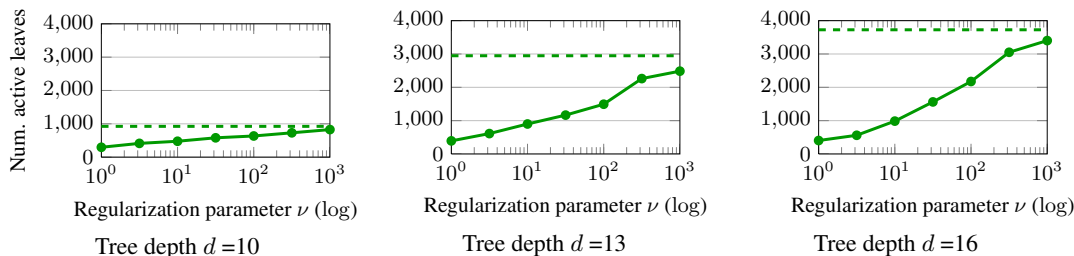

Figure 3: The effect of $\nu$ on the structure of the trees trained by MNIST. A small value of $\nu$ prunes the tree to use far fewer leaves than an axis-aligned baseline used for initialization (dotted line).

greedy trees achieve the best test performance across tree depths across multiple datasets. Secondly, trees trained using our non-greedy approach seem to be less susceptible to overfitting and achieve better generalization performance at various tree depths. As described below, we think that the norm regularization provides a principled way to tune the tightness of the tree's fit to the training data. Finally, the comparison between non-greedy and CO2 [18] trees concentrates on the non-greediness of the algorithm, as it compares our method with its simpler variant, which is applied greedily one node at a time. We find that in most cases, the non-greedy optimization helps by improving upon the results of CO2.

A key hyper-parameter of our method is the regularization constant $\nu$ in (9), which controls the tightness of the upper bound. With a small $\nu$, the norm constraints force the method to choose a $W$ with a large margin at each internal node. The choice of $\nu$ is therefore closely related to the generalization of the learned trees. As shown in Fig. 3, $\nu$ also implicitly controls the degree of pruning of the leaves of the tree during training. We train multiple trees for different values of $\nu \in \{0.1, 1, 4, 10, 43, 100\}$, and we pick the value of $\nu$ that produces the tree with minimum validation error. We also tune the choice of the SGD learning rate, $\eta$, in this step. This $\nu$ and $\eta$ are used to build a tree using the union of both the training and validation sets, which is evaluated on the test set.

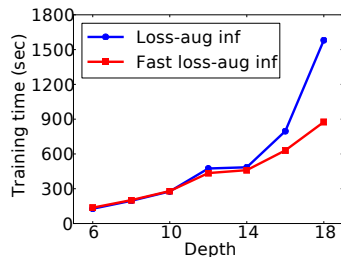

Figure 4: Total time to execute 1000 epochs of SGD on the Connect4 dataset using loss-agumented inference and its fast varient.

To build non-greedy trees, we initially build an axis-aligned tree with split functions that threshold a single feature optimized using conventional procedures that maximize information gain. The axis-aligned split is used to initialize a *greedy* variant of the tree training procedure called CO2 [18]. This provides initial values for $W$ and $\Theta$ for the non-greedy procedure.

Fig. 4 shows an empirical comparison of training time for SGD with loss-augmented inference and fast loss-augmented inference. As expected, run-time of loss-augmented inference exhibits exponential growth with deep trees whereas its fast variant is much more scalable. We expect to see much larger speedup factors for larger datasets. Connect4 only has $55,000$ training points.

## 8 Conclusion

We present a non-greedy method for learning decision trees, using stochastic gradient descent to optimize an upper bound on the empirical loss of the tree's predictions on the training set. Our model poses the global training of decision trees in a well-characterized optimization framework. This makes it simpler to pose extensions that could be considered in future work. Efficiency gains could be achieved by learning sparse split functions via sparsity-inducing regularization on $W$. Further, the core optimization problem permits applying the kernel trick to the linear split parameters $W$, making our overall model applicable to learning higher-order split functions or training decision trees on examples in arbitrary Reproducing Kernel Hilbert Spaces.

**Acknowledgment.** MN was financially supported in part by a Google fellowship. DF was financially supported in part by NSERC Canada and the NCAP program of the CIFAR.

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
