[Supplementary Material]

# Efficient Non-greedy Optimization of Decision Trees and Forests: Supplementary Material

**Mohammad Norouzi**[1]  **Maxwell D. Collins**[2] *  **Matthew Johnson**[3]
**David J. Fleet**[4]  **Pushmeet Kohli**[5]

[1,4] Department of Computer Science, University of Toronto
[2] Department of Computer Science, University of Wisconsin-Madison
[3,5] Microsoft Research

## 1 Proofs

**Upper bound on loss.** For any pair $(\mathbf{x}, y)$, the loss $\ell(\Theta^\mathsf{T} f(\mathrm{sgn}(W\mathbf{x})), y)$ is upper bounded by:

$$\ell(\Theta^\mathsf{T} f(\mathrm{sgn}(W\mathbf{x})), y) \leq \max_{\mathbf{g} \in \mathcal{H}^m} \left\{ \mathbf{g}^\mathsf{T} W\mathbf{x} + \ell(\Theta^\mathsf{T} f(\mathbf{g}), y) \right\} - \max_{\mathbf{h} \in \mathcal{H}^m} \left\{ \mathbf{h}^\mathsf{T} W\mathbf{x} \right\}. \tag{1}$$

*Proof.*

$$
\begin{aligned}
\mathrm{RHS} \;&=\; \max_{\mathbf{g} \in \mathcal{H}^m} \left\{ \mathbf{g}^\mathsf{T} W\mathbf{x} + \ell(\Theta^\mathsf{T} f(\mathbf{g}), y) \right\} - \max_{\mathbf{h} \in \mathcal{H}^m} \left\{ \mathbf{h}^\mathsf{T} W\mathbf{x} \right\} \\
&=\; \max_{\mathbf{g} \in \mathcal{H}^m} \left\{ \mathbf{g}^\mathsf{T} W\mathbf{x} + \ell(\Theta^\mathsf{T} f(\mathbf{g}), y) \right\} - \mathrm{sgn}(W\mathbf{x})^T W\mathbf{x} \\
&\geq\; \max_{\mathbf{g} \in \{\mathrm{sgn}(W\mathbf{x})\}} \left\{ \mathbf{g}^\mathsf{T} W\mathbf{x} + \ell(\Theta^\mathsf{T} f(\mathbf{g}), y) \right\} - \mathrm{sgn}(W\mathbf{x})^T W\mathbf{x} \\
&=\; \mathrm{sgn}(W\mathbf{x})^T W\mathbf{x} + \ell(\Theta^\mathsf{T} f(\mathrm{sgn}(W\mathbf{x})), y) - \mathrm{sgn}(W\mathbf{x})^T W\mathbf{x} \\
&=\; \ell(\Theta^\mathsf{T} f(\mathrm{sgn}(W\mathbf{x})), y) \\
&=\; \mathrm{LHS}
\end{aligned}
$$

$\square$

**Proposition 1.** *The upper bound on the loss becomes tighter as a constant multiple of $W$ gets larger. More formally, for any $\alpha > \beta > 0$, we have:*

$$\max_{\mathbf{g} \in \mathcal{H}^m} \left\{ \alpha \mathbf{g}^\mathsf{T} W\mathbf{x} + \ell(\Theta^\mathsf{T} f(\mathbf{g}), y) \right\} - \max_{\mathbf{h} \in \mathcal{H}^m} \left\{ \alpha \mathbf{h}^\mathsf{T} W\mathbf{x} \right\} \leq$$

$$\max_{\mathbf{g}' \in \mathcal{H}^m} \left\{ \beta {\mathbf{g}'}^\mathsf{T} W\mathbf{x} + \ell(\Theta^\mathsf{T} f(\mathbf{g}'), y) \right\} - \max_{\mathbf{h}' \in \mathcal{H}^m} \left\{ \beta {\mathbf{h}'}^\mathsf{T} W\mathbf{x} \right\}. \tag{2}$$

*Proof.* Let

$$\widehat{\mathbf{g}}_\alpha = \operatorname*{argmax}_{\mathbf{g} \in \mathcal{H}^m} \left\{ \alpha \mathbf{g}^\mathsf{T} W\mathbf{x} + \ell(\Theta^\mathsf{T} f(\mathbf{g}), y) \right\}, \qquad \widehat{\mathbf{g}}_\beta = \operatorname*{argmax}_{\mathbf{g} \in \mathcal{H}^m} \left\{ \beta \, \mathbf{g}^\mathsf{T} W\mathbf{x} + \ell(\Theta^\mathsf{T} f(\mathbf{g}), y) \right\},$$

then we have:

$$\beta \, \widehat{\mathbf{g}}_\alpha^\mathsf{T} W\mathbf{x} + \ell(\Theta^\mathsf{T} f(\widehat{\mathbf{g}}_\alpha), y) \;\leq\; \beta \, \widehat{\mathbf{g}}_\beta^\mathsf{T} W\mathbf{x} + \ell(\Theta^\mathsf{T} f(\widehat{\mathbf{g}}_\beta), y). \tag{3}$$

We also have:

$$\max_{\mathbf{h} \in \mathcal{H}^m} \left\{ \alpha \, \mathbf{h}^\mathsf{T} W\mathbf{x} \right\} = \alpha \, \mathrm{sgn}(W\mathbf{x})^\mathsf{T} W\mathbf{x}, \quad \text{and} \quad \max_{\mathbf{h} \in \mathcal{H}^m} \left\{ \beta \, \mathbf{h}^\mathsf{T} W\mathbf{x} \right\} = \beta \, \mathrm{sgn}(W\mathbf{x})^\mathsf{T} W\mathbf{x}. \tag{4}$$

Moreover,

$$
\begin{aligned}
\widehat{\mathbf{g}}_\alpha^\mathsf{T} W\mathbf{x} &\leq \operatorname{sgn}(W\mathbf{x})^\mathsf{T} W\mathbf{x} \implies \\
(\alpha - \beta)\,\widehat{\mathbf{g}}_\alpha^\mathsf{T} W\mathbf{x} &\leq (\alpha - \beta)\operatorname{sgn}(W\mathbf{x})^\mathsf{T} W\mathbf{x} \implies \\
(\alpha - \beta)\,\widehat{\mathbf{g}}_\alpha^\mathsf{T} W\mathbf{x} - \alpha\operatorname{sgn}(W\mathbf{x})^\mathsf{T} W\mathbf{x} &\leq -\beta\operatorname{sgn}(W\mathbf{x})^\mathsf{T} W\mathbf{x}\,.
\end{aligned}
\tag{5}
$$

Now, summing the two sides of (3) and (5), and using (4), the inequality is proved. $\qquad\square$