[Reviews · NeurIPS 2015]

Submitted by Assigned_Reviewer_1

Edit after feedback: After discussion with the other reviewers and looking at the rebuttal, I still think this is a good paper that deserves publication on the basis of the elegance of the approach, and the fact that it is likely to lead to more research on related approaches. I do agree that the experimental results are a bit disappointing. Still, with stronger results I'd probably rate the paper even higher, so I think that this score is appropriate.
Summary: Very clearly written paper proposes a nonconvex relaxation for global optimization of decision trees with a fixed structure.

Elegant approach, but somewhat disappointing experimental results.

One simple change would be, after using this global optimization, to throw away the last split in each tree and optimize it and the leaf node in the standard way, which might lead to better results.

Also important to evaluate that global optimization should generally lead to more overfitting.

Submitted by Assigned_Reviewer_2

Summary: A method for jointly training all parameters (decision splits and posterior probabilities) of decision trees is proposed. This is an important topic since current methods train trees in a greedy manner, layer by layer. The task is phrased as a single optimization problem which is then upper-bounded and approximated to obtain a tractable formulation.

Quality - The paper is well written but there might be flaws in the reasoning. See below for details. Clarity - The derivation is clean but experiments and their presentation could be improved. Originality - Investigating joint training for decision trees is an important topic Significance - The proposed solution is valuable although there seem to be some shortcomings. See below for details.

Comments: - Flaw in the derivation of the empirical loss bound? At the very least the following argument in l.216f is confusing: `... for all values of $\bg \neq \sgn(W\bx)$, the right hand side can only become larger than when $\bg = \sgn(W\bx)$ ...' Let's look at the following example to illustrate why I'm confused: W\bx = [5;-5]

--> \hat \bh = [1;-1] = \sgn(W\bx)

--> \hat \bh^T W\bx = 10

Choosing g = [1;1] results in g^T W\bx = 0. Therefore the right hand side of Eq. (7) is

g^T W\bx - \hat \bh^T W\bx +

\ell(...) = -10 + \ell(...)

which could possibly be smaller than \ell(...) and hence not a valid upper bound. Note that the proof of Prop. 1 in the supplementary material does not cover this case and is therefore incomplete.

- Why are some of the leaves not assigned any data points? This seems strange and could indicate a problem with the objective or the initialization. The suggested heuristic of a fixed assignment of data-points to tree leaves seems sub-optimal but could possibly be motivated via expectation maximization/concave-convex procedure, similar to other clustering techniques?

- I'm wondering about the importance for the design of the efficient loss-augmented inference. The depth of the tree seems to be at most 16 and greedily checking 2^16 = 65536 values still seems feasible. How much did the restricted search space introduced by the Hamming ball impact performance? One result is given in the supplementary material but does this generalize? And what means `# leaves'?

- I'm missing a baseline which trains non-axis aligned split functions of trees in a standard greedy manner, i.e., without any initialization.

- The depth of the tree needs to be specified ahead of time. I think this limitation should be discussed more carefully. The authors could also provide a more careful experimental evaluation regarding the depth parameter.

--------------- Thanks to the authors for carefully explaining the reasoning behind the loss bound (Eq. 7). You might want to replace \arg\max with \max. As stated in my summary I adjusted my score.

An experiment comparing the runtime could strengthen the paper. A more careful evaluation of the method would be desirable for the reader.
Summary: The submission considers an interesting problem of jointly training all parameters of decision trees. The experimental evaluation is somewhat limited and there could potentially be a flaw in the derivation. I'm happy to adjust my score given a careful explanation in the rebuttal.

Submitted by Assigned_Reviewer_3

The authors propose an interesting way to search for the best decision tree of a specific depth by posing it as a structured prediction problem. The methods work better than typical greedy approach for decision tree building in terms of test set accuracy, but the improvements tend to be small.

Summary: The authors propose an interesting way to search for the best decision tree of a specific depth by posing it as a structured prediction problem. The methods work better than typical greedy approach for decision tree building in terms of test set accuracy, but the improvements tend to be small.

Submitted by Assigned_Reviewer_4

The submission proposes an algorithm for global optimization of decision trees based on a reformulation of the problem as (latent) structured prediction. The latent structured variables are sign vectors encoding the decisions made at each internal node in the tree.

A non-differentiable and non-convex loss is proposed, which is then approximated by a differentiable (but still non-convex) upper bound.

Minimizing this loss via SGD entails solving a sequence of loss-augmented inference problems, for which efficient algorithms are given. Experiments are shown comparing this method to a standard greedy training method.

The key contributions of the work are casting the global DT optimization as structured prediction and devising efficient algorithms to optimize the objective.

The way this is done is novel and interesting, and raises interesting questions.

I have some concerns, however, about some practical considerations-especially concerning the motivation for the method and how significant the performance gains over naive methods are.

Considered purely as a method to globally optimize DTs, the method is intriguing in several ways.

The first advantage gained by casting the optimization as structured prediction is being able to perform SGD by solving a sequence of loss-augmented optimization problems that leverage the special structure of the problem.

However, I think the most interesting result of this approach is that the gradient updates are sparse-if I'm not mistaken, each gradient step involves changing only one node's weight vector.

The submission presents this mainly as a computational advantage, but I believe that this property raises more intriguing possibilities that are not really acknowledged in the paper.

One could imagine optimizing over an infinitely deep tree by initially setting all weights to zero, and then gradually growing the tree by making more weights nonzero, thus letting the model's complexity grow naturally to fit the data.

This raises interesting questions: for example, would such a method prove to be equivalent to, or a variation of, any known greedy strategy?

This may be the case, if it turns out that it is always preferable to introduce a new node rather than revise an old node's weights.

Or, with appropriate regularization, would such a method instead converge to a tree with a limited number of nodes (as Fig. 2 might suggest)?

I think this issue cuts to the core of what makes the proposed method worthwhile compared to naive DTs, and deserves perhaps more careful analysis than is currently offered by the paper.

For instance, a critical issue left ambiguous is this: what happens if some node's weight is equal to zero?

The discussion on page 3 implies that we take the right branch, which seems like a dubious choice.

It seems like it would be preferable to output a decision at this node; later, we might decide to make this weight nonzero, corresponding to splitting the node.

Precisely defining what happens here is important, because it has a direct analog to pruning strategies for greedy training.

I have one significant practical concern regarding the motivation for the work.

Namely, I would say that plain DTs are useful mainly to the extent that they are interpretable; if the interpretability requirement is dropped, then it is usually better to use other methods, which generally offer much better performance.

As stated, I would expect the method to produce non-interpretable results, since the L2 regularization would produce dense weight vectors.

Although switching to L1 regularization might help, it is possible that performance would drop if we were forced to regularize to the point where exactly 1-sparse splits were produced.

So, this would definitely need to be tested.

My other main concern is that the experiments don't go into enough detail regarding other baselines.

No details are given as to which pruning strategies were employed for the naive approach.

Since a range of regularization parameters were tried for the proposed method, I would expect the naive method to be similarly tried with a few different pruning strategies and/or parameters.

I think it would also be fair to try some other trivial strategies for global optimization, such as training a greedy method with random subsets of the training data, and choosing the one that minimizes the desired loss on the full training or validation set.

Regarding clarity, the paper is reasonably well-written. My only comment is that the parts describing inference could be a bit clearer.

Section 6.2, for example, could be summarized by saying that the objective is first maximized for all g corresponding to a given leaf, which is then maximized over all leaves.

Section 6.4 could also be a little more explicit.

In summary, I think this is a very clever approach that raises some very interesting questions.

However, it is unclear whether this is a practically useful method at this point, both due to the aforementioned interpretability issue (why use plain DTs in the first place, if they are not interpretable?) and due to a lack of details in the experiments.

PS: although the title mentions forests, the paper does not address this case.

POST-REBUTTAL COMMENTS

I still think that the case of zero weights is a critical case that deserves further analysis, for the reasons I brought up in my review.

Reading the rebuttal, it sounds like the weights are probably initialized randomly, which is why this issue doesn't seem to come up in practice.

However, I have a feeling that what would really make this method interesting would be the case where the weights are initially zero and are gradually increased.

I encourage the authors to consider this direction.
Summary: The submission proposes a clever reformulation of global optimization of decision trees as structured prediction, along with an efficient algorithm to solve the optimization. Some aspects of the motivation for the work and experimental results are not that convincing, however.

Submitted by Assigned_Reviewer_5

This paper presents an algorithm for learning a decision tree with linear split functions at the nodes. The algorithm optimizes the tree parameters (i.e. the weights in the split functions and the distribution of the output variable at each leaf) in a global way, as opposed to the standard greedy way of learning trees. An upper bound on the tree's empirical loss is proposed, and to achieve a smoother optimization problem, this upper bound is regularized, via a L2 regularization of the split function weights. The algorithm then consists in using a stochastic gradient descent in order to identify the tree parameters that minimize the upper bound.

The tree optimization problem is elegantly formulated, by establishing a link with structured prediction with latent variables. However, the empirical results that are shown in the paper do not really convince that a global optimization is a better alternative to a greedy learning of the tree. The globally optimized tree outperforms significantly the greedy tree on a couple of datasets only, while being more intensive computationally.

The paper is well structured, but some points need to be clarified:

- The stable version of SGD is not described very clearly. How are data points assigned to leaves? It would probably help to have a pseudo-code.

- Figure 2: are these results obtained when applying SGD or stable SGD? What does an "active" leaf mean exactly?

- In line 10 of Algorithm 1, \Theta is updated and then projected on a simplex. In practice, how do you solve this projection problem? It would also maybe be worth mentioning that this projection ensures that the each line of \Theta sums up to one.

Minor comments / typos:

- In lines 407-408, you say that you tune the "number of features evaluated when building an ordinary axis-aligned decision tree". I am not sure to understand, since all the features are evaluated at each node when building a standard decision tree. - In equations (7), (8) and (9), "argmax" must be replaced with "max". - Figure 1 is never mentioned in the main text. - Line 141: "...to the index of the leaf on by this path." Something is wrong at the end of this sentence. - Line 238: "...the solution to loss-augmented inference..." -> the solutions to... - Line 301: "An key observation..." -> A key... - Line 360: What is FSGD? - Line 404: "...a tree with minimum training errors." I suppose you mean "tuning" errors?

In the supplementary material:

- Line 29: max_{g \in sgn...} -> max_{g = sgn...} - Line 30: sgn(Wx)^T Wx + l(\Theta^Tf(g), y)... -> sgn(Wx)^T Wx +

l(\Theta^Tf(sgn(Wx)), y)

Summary: The tree optimization problem is elegantly formulated, but the results shown are not convincing.

Author Feedback
Author rebuttal: We thank reviewers for their feedback.

R1: Flaw in the derivation of the empirical loss bound?

There is no flaw in the proof of Eq (7). Supplementary material includes a step by step algebraic proof of the inequality. In the proof sketch in Lines 215-218, we do not argue that any assignment of binary vectors to g increases value of (gWx + loss), but, we argue that the value of max(gWx + loss) is always as large or larger than gWx + loss when g = sgn(Wx), because of the max operation. We will make the description clearer in the final version.

R1: Why are some of the leaves not assigned any data points?

During SGD, we may reach a set of parameters such that none of the training data pass the binary tests to reach a specific leaf. We can re-initialize the weights to activate such leaves again, but we think of this property of the algorithm as an automatic pruning procedure that arises naturally out of our formation, and is helpful for generalization.

R1: importance of the efficient loss-augmented inference

For trees of depth d, basic loss-augmented inference requires O(2^d) operations, whereas our faster version requires O(d^2) operations. For d=16, this results in a ~256 fold improvement in speed. Experiments reveal no significant drop in accuracy (last fig in supp). We will include a run-time comparison in the final version.

R1: I'm missing a baseline which trains non-axis aligned split functions of trees in a standard greedy manner

We can consider additional baselines that greedily compute oblique splits. A standard method is to pick the best from randomly sampled coefficient vectors. We find this not competetive on all but the lowest-dimensional datasets.

Running experiment in Fig 2, the best training and test accuracy at depth 20:

connect4 0.71 0.67
pendigits 0.95 0.95
protein 0.50 0.43
mnist 0.80 0.79
letter 0.83 0.83
sensit 0.76 0.73
segment 0.95 0.94

R1: The depth of the tree needs to be specified..

Tree induction algorithms also require one of the following to be specified: the depth of the tree, the number of leaves, or the minimum number of training instances per leaf. We can select the best tree depth by cross-validation.

R2: Lack of interpretability, sparse regularization

Oblique splits can also be visualized in different ways, eg, split weights can be viewed as a matched filter in vision applications. We respectfully disagree that decision trees are only useful because of interpretability, as for some moderate-size datasets they offer state-of-the-art results. Our approach can also be used to learn axis aligned splits through use of sparse regularizers.

R2: what happens if some node's weight is zero?

It is extremely unlikely that a node's weights become zero. In this case we can select either of the two subtrees to proceed, but with datasets we've used this has not happened.

R2: lack of details in experiments

We will improve the description of experiments, and we will show two additional baselines: randomized oblique trees and OC1.

R3: Stable SGD is not described very clearly. How are data points assigned to leaves?

The data points are assigned to leaves given the parameters at the beginning of each stable SGD iteration. The leaf assignments are kept fixed within several SGD updates. We will include pseudo-code in the supplementary material.

R3: Fig 2: are these results obtained when applying SGD or stable SGD? What does an "active" leaf mean exactly?

These results are obtained by applying stable SGD. An active leaf is a leaf to which at least one training datum is assigned.

R3: line 10 of Algorithm 1, \Theta is updated and then projected on a simplex.. how do you solve this projection problem?

In our formulation, \theta is exponentiated and normalized through softmax to produce a probability distribution (see Eq. (3)). Thus, no explicit projection onto the simplex is required. Line 10 of Alg 1 includes a projection for general cases in which a projection may be necessary.

R3: What is FSGD?

SGD with fast loss-augmented inference.

R4: non-greedy algorithms for decision tree optimization

Approaches by Bennet [1-3] are computationally expensive and limited by tree depth. To our knowledge, no non-greedy baseline exists in the literature that applies to moderate-sized trees and thousands of training examples.

R4: comparisons to neural net

These are standard datasets where results of other models are known. Our focus was to compare against decision tree algorithms.

R5: improvements tend to be small.

In two of datasets, the improvements are relatively large, especially when tree depth is small. In two other datasets the improvements are small but positive, outperforming axis-aligned splits. That said, our main contribution is a new formulation connecting decision trees and latent structural SVMs. This formulation can also be used for training other related models such as mixtures of experts.